# Comparative Analysis of Laparoscopic and Robotic Transperitoneal Adrenalectomy Performed at a Single Institution

**DOI:** 10.3390/medicina58121747

**Published:** 2022-11-29

**Authors:** Yun Suk Choi, Ji Sun Lee, Jin Wook Yi

**Affiliations:** Department of Surgery, Inha University Hospital, College of Medicine, Incheon 22332, Republic of Korea

**Keywords:** robotic adrenalectomy, minimally invasive surgery, adrenal gland neoplasm, laparoscopic adrenalectomy

## Abstract

*Background and Objectives:* Laparoscopic adrenalectomy (LA) is the standard surgical approach for adrenalectomy. At present, robotic adrenalectomy (RA) has been introduced at various hospitals. This study evaluated our initial experience with robotic adrenalectomy compared with conventional laparoscopic adrenalectomy. *Materials and Methods:* From October 2018 to March 2022, 56 adrenalectomies were performed by a single endocrine surgeon. Thirty-two patients underwent LA (LA group), and twenty-four patients underwent RA (RA group). *Results:* Patients in the RA group were significantly younger than those in the LA group (48.6 ± 9.7 years vs. 55.1 ± 11.4 years, *p* = 0.013). The RA group had a shorter operation time than the LA group (76.1 ± 28.2 min vs. 118.0 ± 54.3 min, *p* < 0.001). The length of hospital stay and postoperative pain level between the two groups were similar. There were no complications in the RA group. There was no significant difference in the pathologic diagnosis between the two groups. The cost of surgery was significantly higher in the RA group than in the LA group (5288.5 US dollars vs. 441.5 ± 136.8 US dollars, *p* < 0.001). *Conclusions:* In our initial experience, RA showed a shorter operation time than LA and no complications. RA could be a viable alternative surgical option for adrenalectomy, notwithstanding its higher cost.

## 1. Introduction

The adrenal glands are mustard-colored paired organs located in the superomedial part of the kidney in the retroperitoneum. They secrete various kinds of steroid hormones, such as cortisol and aldosterone from the cortex and catecholamine from the medulla. An adrenalectomy is recommended for malignancies such as adrenocortical cancer, size-increasing adrenal tumors, and functional adrenal tumors such as those causing primary aldosteronism, Cushing’s syndrome, and pheochromocytoma [1].

Adrenal surgery has a long history with steady progress. The first adrenal surgery was performed by Thorton in 1889, wherein an adrenal mass was described as a large sarcoma with a left suprarenal capsule [2]. In 1914, the first planned adrenalectomy was performed by Perry Sargent [3]. The first flank approach for a pheochromocytoma was performed by Charles Mayo in 1927 [4]. Most early adrenal surgeries were conducted to remove large tumors, and the incisions were essentially similar to those made for renal surgery. In adrenal surgery, the open method was the only surgical option until 1992 [5].

The first laparoscopic adrenalectomy (LA) using a lateral transperitoneal approach was reported in 1992 [6]. Another method of LA using a posterior retroperitoneal approach was introduced in 1995 [7]. Currently, LA is considered the gold standard for the excision of small, benign functional adrenal tumors [8]. With the emergence of robotic technology, robot-assisted adrenal gland surgery was first reported in 1999, and the first robotic adrenalectomy (RA) was reported in 2001 [9,10]. RA has gradually become popular in many countries, and the proportion of RAs has gradually been increasing [11,12].

The da Vinci robotic surgical device facilitates the use of many advanced techniques including high-quality three-dimensional (3D) vision and intuitive controlled movement with seven degrees of freedom through the endowrist function. Surgeons can perform surgery more comfortably and delicately using this robotic device, which can lead to better surgical results than conventional endoscopic surgery [13]. Robotic surgery may be useful in adrenalectomy, which involves precise movements in a limited space. However, research on RA suffers from a lack of cases compared with surgery on other organs [14,15].

Our hospital started performing robotic surgery using da Vinci Xi in 2018, and RA has also been performed. This study evaluates the initial experiences with RA compared with conventional LA performed by a single endocrine surgeon. 

## 2. Methods

### 2.1. Study Design

We analyzed the electronic medical records of patients who underwent either LA or RA at Inha University Hospital. The potential surgical candidates for transperitoneal RA were the same as those for LA, and RA was chosen when the patient agreed to undergo surgery using this technique despite the greater expense than the laparoscopic approach. We reviewed clinical information data including age, gender, body mass index (BMI), preoperative clinical diagnosis, tumor location and size, type of combined operation, final pathology diagnosis, operation time, estimated blood loss volume, postoperative pain, postoperative length of hospital stay, complications, and surgical cost. All surgeries were recorded, and the videos were reviewed. We captured important scenes and calculated the actual surgery time based on the surgery videos. In the case of combined surgery, the operation time and cost of surgery were only evaluated for the adrenalectomy. The cost of the surgery was calculated as the fees for surgery only, excluding extra expenses including hospitalization and other expenses. 

### 2.2. Patients

From October 2018 to March 2022, 62 adrenalectomies were performed by a single endocrine surgeon (JW Yi) at Inha University Hospital, Incheon, South Korea. All patients underwent adrenalectomy using a transperineal approach. Two groups were formed: the LA group and the RA group. We excluded three cases of open conversion (only occurred in LA) and three cases of initial open surgery. A total of 56 patients (24 patients in the RA group and 32 in the LA group) were included in the analysis.

### 2.3. Statistics and Ethical Considerations

All statistical analyses were performed using IBM SPSS Statistics 28 (IBM Corporation, Armonk, NY, USA). Continuous variables are presented using the mean ± standard deviation. Unpaired *t*-tests were used to compare the means. The chi-square test or Fisher’s exact test was applied to the cross-table analysis, depending on the sample size. 

The ethics of this study were approved by the institutional review board of the author’s institution (INHAUH 2022-05-015). 

### 2.4. Operative Procedure for RA

LA was performed via a traditional lateral transperitoneal approach, as previously reported elsewhere [16]. Our procedure for transperitoneal RA is described below. 

Under general anesthesia, the patient was placed in the lateral decubitus position. The patient’s bed was bent at an angle of 80° to expose the side as much as possible. Figure 1 shows the trocar placement in RA. For the right side, three robotic arms and an additional 5 mm port were required for liver mobilization, as shown in Figure 1A. For the left side, only three robotic arms were needed, as shown in Figure 1B. For effective movement of the robotic endowrist in the abdominal cavity, robot trocars were placed 3 cm apart from the subcostal margin to enable the unrestricted movement of the robotic arms. To prevent collision of the robot arms, a distance of approximately 5 cm was maintained between the trocars. We used three types of robotic endowrist instruments: prograsp forceps, long bipolar, and vessel sealer extend. 

The steps for right RA are shown in Figure 2. The right triangular ligament was detached to sufficiently mobilize the right liver (Figure 2A). Then, a snake retractor was inserted, the mobilized right liver pulled up, and the location of the inferior vena cava (IVC) was identified (Figure 2B). After opening Gerota’s fascia, the upper pole of right kidney was identified. After dissecting upward along the superior pole of the right kidney, the right adrenal gland was identified (Figure 2C). The medial side of the adrenal gland was carefully dissected to find the adrenal vein (Figure 2D), and it was divided after hem-o-lock ligation (Figure 2E). The adrenal resection was terminated by completing the peripheral adrenal dissection (Figure 2F). 

The steps for left RA are shown in Figure 3. The left colon was lowered by dissecting around the splenic flexure (Figure 3A). Then, the spleen was pulled downward by dissecting the splenocolic and splenophrenic ligaments (Figure 3B). After dissection around the distal pancreas and Gerota’s fascia, the location of the adrenal gland was revealed (Figure 3C). The superior pole of the left kidney was exposed, and careful dissection was performed around the adrenal gland (Figure 3D). After identifying the adrenal vein, it was ligated by hem-o-lock and cut (Figure 3E). The remaining soft tissue around the adrenal gland was detached, and the adrenalectomy was completed (Figure 3F).

## 3. Results

The clinical characteristics of the RA and LA groups are summarized in Table 1. The mean age in the RA group was significantly lower than that in the LA group (48.6 ± 9.7 years vs. 56.1 ± 11.4 years, *p* = 0.013). Gender and BMI were not different between the two groups. The clinical diagnoses included primary aldosteronism, Cushing syndrome, and the increasing size of a non-functioning adenoma. There were no significant differences in the tumor location and size between the two groups. The RA group had four combined operation cases with a robotic approach including cholecystectomy, partial nephrectomy, and total hysterectomy with bilateral salpingo-oophorectomy. The LA group had one combined operation case, i.e., laparoscopic cholecystectomy.

Table 2 summarizes the surgical and clinical outcomes of the RA and LA groups. The RA group exhibited a shorter operation time than the LA group (76.1 ± 28.2 min vs. 118.0 ± 54.3 min, *p* < 0.001). There were no significant differences in the estimated blood loss, hospital stay after surgery, and visual analogue pain scale on postoperative days 1 and 2 between the two groups. There were no postoperative complications in the RA group, while a port-site hernia occurred in a single case in the LA group. The cost of surgery was significantly higher in the RA group than in the LA group (5288 US dollars vs. 441.5 ± 136.8 US dollars, *p* < 0.001). The pathologic diagnoses included adrenal cortical adenoma, pheochromocytoma/paraganglioma, and myelolipoma. There was no significant difference in the pathologic diagnosis between the two groups.

Table 3 summarizes the results of other studies in which RAs were performed. In our study, the tumor size was similar to that in previous studies, but the operation time was reported to be shorter than that in previous studies.

## 4. Discussion

Compared with conventional open surgery, minimally invasive surgery (MIS) has many advantages: it is not only cosmetically superior but also results in less postoperative pain, a shorter hospital stay, and favorable oncologic outcomes in cancer surgery [39,40,41]. MIS has been mainly performed using a laparoscopic approach. With the development of robotic surgical systems, MIS using a robotic approach has been established, and many clinical studies have been conducted [42]. 

LA allows for better access to adrenal glands than open surgery. The pneumoperitoneum enables a wider surgical space into which other organs can easily be retracted in a downward direction. Dissection around the adrenal glands is also easier because of the fine and straightforward laparoscopic instruments. The adrenal vessels can be easily ligated using a laparoscopic clip applier compared with open adrenalectomy [43,44]. The size of the incision is very small, and only three or four incisions are required for trocar placement compared with open methods. Given these advantages, LA is considered the standard method [8]. 

The first robotic adrenal surgery was reported in 1999 [9]. Currently, several clinical experiences of robotic adrenalectomy are being reported in various hospitals as described in Table 3. Some studies that compared RA to the laparoscopic approach suggested that there are no specific advantages of RA [21,23,24,27,29,32,34,35]. However, we found that the operation time was significantly shorter in the RA group, as described in Table 1. Furthermore, there were no postoperative complications in the RA group. We suggest that the reason for these results may be that the surgeon who performed RA at our hospital had considerable experience in robotic surgery, i.e., had performed more than 500 robotic thyroid surgeries. There was no need for adaptation or a learning curve to perform robotic surgery. In addition, he had sufficient LA experience.

In our study, there was one case of a port-site hernia in the LA group. It occurred two months after surgery through the 12 mm port site and was corrected by surgical treatment. This may have been due to surgical failure to close the 12 mm port site. As 5 mm and 8 mm trocars are used in robot adrenalectomy, it may be helpful to prevent port-site hernia. We performed two cases of right RA with cholecystectomy. The lateral decubitus position is used in adrenalectomy, whereas the supine position is used in cholecystectomy. We safely performed both surgeries in a single stage using the lateral decubitus position without changing the position. We believe that right RA and cholecystectomy can be performed safely in a single stage using the lateral decubitus position without any special position changes.

The advantages of the robotic system in an adrenalectomy are as follows. First, there is considerable extracorporeal and intracorporeal fighting in LA because the surgeon and assistant are too close during surgery. In contrast, RA does not involve such collision, and more free surgical movement is possible. Second, more and precise free movement is possible by using the endowrist of the robot arm than by using the laparoscopic device. This is very convenient for liver mobilization and enables increased mobilization. In LA, there is also a risk of surrounding structure injury because the angle of the laparoscopic arm is not parallel to the IVC. The angulation of the robotic arm makes this possible and reduces this risk. Third, the 3D augmented view of the robotic system provides a very clear and accurate view to the operator, and the operator can directly control it to obtain the desired view. 

The disadvantages of RA are as follows: First, it does not provide tactile sense from the instrument to the surgeon’s hand. This makes it difficult for inexperienced surgeons to distinguish tissues or organs. Second, RA is more than five times as expensive as LA. Many people have a personal medical insurance system in South Korea, and patients pay only 0–20% of the surgical cost. In this case, there is no difference between robotic surgery and laparoscopic surgery. RA showed some advantages compared to LA in our study. RA can be recommended to patients with personal medical insurance. The final concern is proper training for surgeons. Although robotic surgery is popular worldwide, there are few institutions that perform RA on a large scale. This makes it difficult for surgeons to learn about robotic adrenal surgery. 

One limitation of our study is that RA is not yet a widely used surgical technique in South Korea; thus, there are not enough surgical cases. There are not many indications for adrenalectomy, and there are not many institutions that perform adrenalectomy in South Korea. Nevertheless, it is meaningful to analyze 56 cases over approximately two years. As more cases are accumulated in the future, additional analyses will be conducted. 

## 5. Conclusions

The transperitoneal RA is a promising surgical method for adrenalectomy. RA has many advantages over conventional LA. The operation time of RA was shorter than that of LA, and no significant complications occurred. 

Several studies have reported their experiences with adrenalectomy using a single-port robot system [17,44]. We will adopt a new single-port robot system in the near future and evaluate the advantages and disadvantages of this new robot system. 

## Figures and Tables

**Figure 1 medicina-58-01747-f001:**
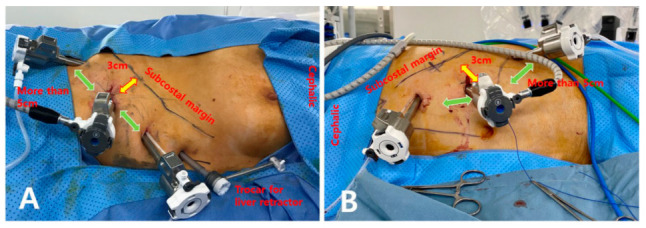
Trocar sites for robotic adrenalectomy. (**A**) Right adrenalectomy. (**B**) Left adrenalectomy.

**Figure 2 medicina-58-01747-f002:**
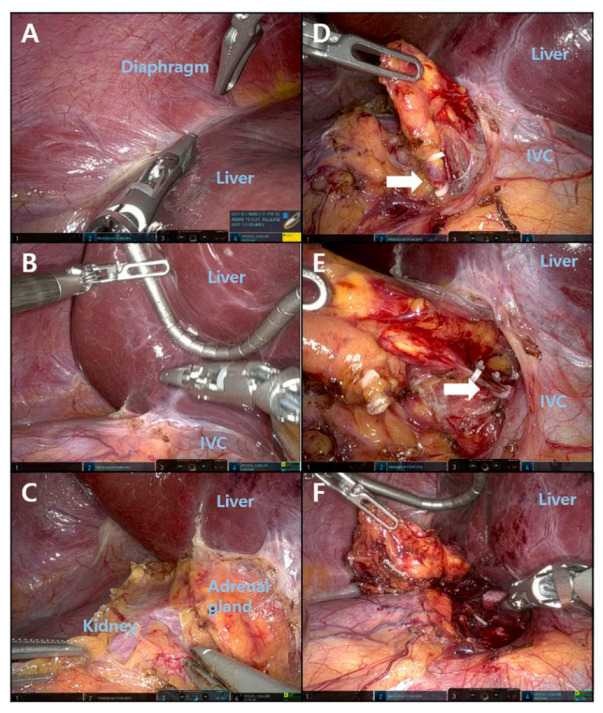
Procedure for robotic right adrenalectomy. (**A**) Liver mobilization. (**B**) Liver retraction and inferior vena cava identification. (**C**) Gerota’s fascia dissection and adrenal gland identification. (**D**) Adrenal vein identification. (**E**). Clipping of the adrenal vein. (**F**) Adrenalectomy completion.

**Figure 3 medicina-58-01747-f003:**
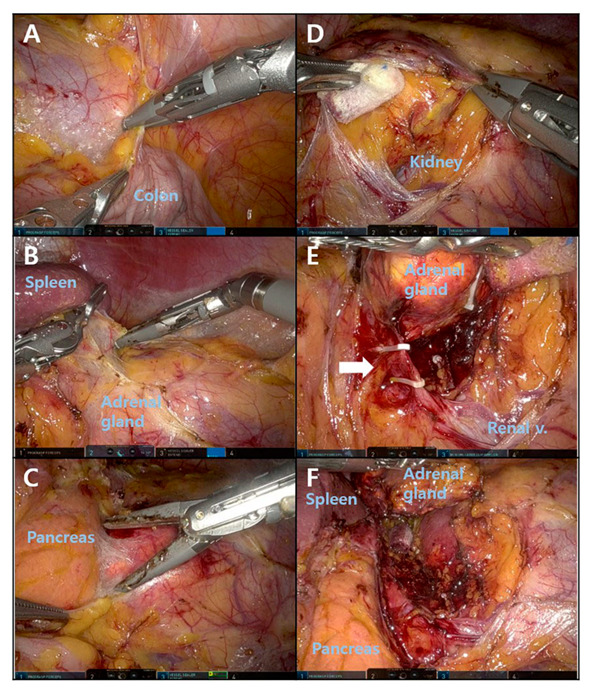
Procedure for robotic left adrenalectomy. (**A**) Left colon mobilization. (**B**) Spleen mobilization. (**C**) Pancreas and Gerota’s fascia dissection. (**D**) Exposure of the left kidney. (**E**) Adrenal vein identification and ligation. (**F**) Adrenalectomy completion.

**Table 1 medicina-58-01747-t001:** Clinical characteristics of patients undergoing robotic adrenalectomy and laparoscopic adrenalectomy.

Variable	Total (*n* = 56)	RA (*n* = 24)	LA (*n* = 32)	*p*-Value
Age (years, mean ± SD)	52.9 ± 11.2 (25–76)	48.6 ± 9.7 (29–71)	56.1 ± 11.4 (25–76)	0.013
Gender				
Male	28 (50%)	12 (21%)	16 (29%)	1.000
Female	28 (50%)	12 (21%)	16 (29%)	
BMI ^a^ (kg/m^2^, mean ± SD)	25.9 ± 4.4 (17.3–35.0)	26.6 ± 4.9 (18.2–35.0)	25.4 ± 4.1 (17.3–32.7)	0.344
Clinical diagnosis				
Primary aldosteronism	16 (28.6%)	9 (37.5%)	7 (21.9%)	
Cushing’s syndrome	14 (25.0%)	6 (25.0%)	8 (25%)	
Non-functioning Adenoma	12 (21.4%)	3 (12.5%)	9 (28.1%)	
PPGLs ^b^	9 (16.1%)	5 (20.8%)	4 (12.5%)	
r/o Metastatic mass	3 (5.4%)	1 (4.2%)	2 (6.25%)	
Adrenal cortical cancer	2 (3.6%)	0 (0.0%)	2 (6.25%)	
Tumor location				
Right	27 (48%)	14 (25%)	13 (23%)	0.196
Left	29 (52%)	10 (18%)	19 (34%)	
Tumor size (cm, mean ± SD)	3.5 ± 2.2 (1.3–9.6)	3.1 ± 1.8 (1.3–8.0)	3.8 ± 2.4 (1.3–9.6)	0.269
Combined operation	3			
Cholecystectomy	2	2	1	
Partial nephrectomy	1	2	0	
TH + BSO ^c^	3	1	0	

^a^ Body mass index, ^b^ Pheochromocytoma/paraganglioma ^c^ Total hysterectomy + bilateral salpingo-oophorectomy.

**Table 2 medicina-58-01747-t002:** Surgical and clinical outcomes of robotic adrenalectomy and laparoscopic adrenalectomy.

Variable	Total (*n* = 56)	RA (*n* = 24)	LA (*n* = 32)	*p*-Value
Operation time (minute, mean ± SD)	100.0 ± 48.6 (46–295)	76.1 ± 28.2 (46–140)	118.0 ± 54.3 (60–295)	<0.001
Estimated blood loss (mL, mean ± SD)	152.1 ± 132.7 (0–400)	164.6 ± 144.8 (0–400)	142.8 ± 124.5 (0–400)	0.548
Hospital stay after operation (days, mean ± SD)	4.0 ± 1.9 (2–9)	4.3 ± 2.1 (2–9)	3.8 ± 1.7 (2–9)	0.326
Visual analog scale				
Postoperative day 1 (0–10, mean ± SD)	2.9 ± 0.6 (0–5)	2.9 ± 0.9 (0–5)	2.9 ± 0.4 (2–4)	0.737
Postoperative day 2 (0–10, mean ± SD)	2.5 ± 0.7 (0–3)	2.5 ± 0.8 (0–3)	2.5 ± 0.7 (1–3)	0.873
Complication				
Port-site hernia	1 (1.8%)	0 (0.0%)	1 (3.1%)	0.382
Cost of surgery (US dollar ^c^, mean ± SD)	2528.8 ± 2422.5 (352.8–5288.5)	5288.5	441.5 ± 136.8 (352.8–1167.1)	<0.001
Pathologic diagnosis				
Adrenal cortical adenoma	35 (62.5%)	16 (66.7%)	19 (59.4%)	
PPGLs ^a^	6 (10.7%)	5 (20.8%)	1 (3.1%)	
Myelolipoma	5 (8.9%)	2 (3.6%)	3 (9.4%)	
Adrenal cortical cancer	2 (3.6%)	0 (0.0%)	2 (6.3%)	
Others ^b^	8 (14.3%)	1 (4.2%)	7 (21.9%)	

^a^ Pheochromocytoma/paraganglioma, ^b^ Adenomatoid tumor, Cortical nodular hyperplasia, Ganglioneuroma, Lymphangioma, Macronodular hyperplasia, Malignant pheochromocytoma, Metastatic Hepatocellular cell, Pseudocys, ^c^ Exchange rate as of 17 April 2022.

**Table 3 medicina-58-01747-t003:** Case series for robotic adrenalectomy.

Year		Study Design	Patient (*n*)	Tumor Size (cm)	Operation Time (min)
2006	Winter [17]	Case series	30	2.4	185
2008	Brunaaud [18]	Case series	100	2.9	171
2011	Giulianotti [19]	Case series	42	5.5	118
2011	Nordenström [20]	Case series	100	5.3	113
2012	Agcaoglu [21]	Comparative	31	3.1	163.2
2012	D’Annibale [22]	Case series	30	5.1	200
2013	Aksoy [23]	Comparative	42	4.0	186
2013	Aliyev [24]	Comparative	26	-	149
2014	Brandao [25]	Comparative	30	3	120
2016	Lee [26]	Case series	33	-	234
2016	Morelli [27]	Comparative	41	-	177
2019	Greilsamer [28]	Case series	303	3.6	89
2019	Kim [29]	Comparative	61	3.7	138
2020	Carmela [30]	Comparative	12	-	93.3
2020	Fu [31]	Comparative	19	8	166.3
2020	Changwei [32]	Comparative	87	4.7	136.1
2020	Ozdemir [33]	Case series	111	3.9	135.4
2020	Fang [34]	Comparative	41	6.2	210.4
2021	Piccoli [35]	Comparative	76	4.0	100.3
2022	Knežević [36]	Case series	12	-	165.1
2022	Erdemir [37]	Case series	30	8.3	194.9
2022	Al-Thani [38]	Comparative	76	4.8	174

## Data Availability

No new data were created or analyzed in this study. Data sharing is not applicable to this article.

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
