# Peer review of "Comparative Analysis of Laparoscopic and Robotic Transperitoneal Adrenalectomy Performed at a Single Institution"

_medicina, 2022, doi:10.3390/medicina58121747_

Round 1
Reviewer 1 Report
This manuscript could easily rise to the level of being published. In its current form, it has a number of deficiencies, which are not difficult to correct:
• - the results chapter must be completed.
· - the discussion chapter must be improved.
· -the correctness of citing the bibliography in the text, in order, must be checked.
The final score places the manuscript in the minor revision needed category.
Author Response
Dear editor of MEDICINA
Sincere thanks for your kind review. According to your opinion, we have added information about the surgical cost of robot adrenalectomy to the discussion section. In addition, the correction for the reference was implemented again.
Revised contents is indicated in red text.
The disadvantages of RA are as follows. First, it does not provide tactile sense from the instrument to the surgeon’s hand. This makes it difficult for inexperienced surgeons to distinguish tissues or organs. Second, RA is more than five times as expensive as LA. Many people have a personal medical insurance system in South Korea, and patients pay only 0-20% of the surgical cost. In this case, there is no difference between robotic surgery and laparoscopic surgery. RA showed some advantages compared to LA in our study. RA can be recommended to patients with personal medical insurance. The final concern is proper training for surgeons. Although robotic surgery is popular worldwide, there are few institutions that perform RA on a large scale. This makes it difficult for surgeons to learn about robotic adrenal surgery.
Your review has been very helpful to our research.
Once again sincerely thank you.

Reviewer 2 Report
1. References quoted in Table 3 need to rearranged , as it has been depicted ahead of discussion . Reference in discussion starts with 17 and in results it starts with 32.
2.Cost effectiveness about RA must also to be included in conclusion.
3. Line no 215- 219- limitation of study - need to rephrase - this is not a recent technique , already there are meta-analysis published on this topic quoting the clinical advantages of RA over LA
Author Response
Dear editor of MEDICINA
Sincere thanks for your kind review. According to your opinion, we have added information about the surgical cost of robot adrenalectomy to the discussion section. And we revised about RA as your recommend. In addition, the correction for the reference was implemented again.
Revised contents is indicated in red text.
The disadvantages of RA are as follows. First, it does not provide tactile sense from the instrument to the surgeon’s hand. This makes it difficult for inexperienced surgeons to distinguish tissues or organs. Second, RA is more than five times as expensive as LA. Many people have a personal medical insurance system in South Korea, and patients pay only 0-20% of the surgical cost. In this case, there is no difference between robotic surgery and laparoscopic surgery. RA showed some advantages compared to LA in our study. RA can be recommended to patients with personal medical insurance. The final concern is proper training for surgeons. Although robotic surgery is popular worldwide, there are few institutions that perform RA on a large scale. This makes it difficult for surgeons to learn about robotic adrenal surgery.
One limitation of our study is that RA is not yet a widely used surgical technique in South Korea; thus, there are not enough surgical cases. There are not many indications for adrenalectomy, and there are not many institutions that perform adrenalectomy in South Korea. Nevertheless, it is meaningful to analyze 56 cases over approximately two years. As more cases are accumulated in the future, additional analyses will be conducted.
Your review has been very helpful to our research.
Once again sincerely thank you.
